# “You Need to Get Over the Difficulties and Stand Up Again”—A Qualitative Inquiry into Young Nurses’ Coping with Lateral Violence from the Feminist Perspective

**DOI:** 10.3390/ijerph18137167

**Published:** 2021-07-04

**Authors:** Aimei Mao, Hon Lon Tam, Pak Leng Cheong, Iat Kio Van

**Affiliations:** Education Department, Kiang Wu Nursing College of Macau, Macau 999078, China; alantam@kwnc.edu.mo (H.L.T.); joecheong@kwnc.edu.mo (P.L.C.); van@kwnc.edu.mo (I.K.V.)

**Keywords:** lateral violence, nursing students, novice nurses, feminist perspective, qualitative research

## Abstract

Previous studies have reported lateral violence (LV) styles among nurses and the adverse impacts of LV on nurses and nursing. Young nurses, including nursing students and novice nurses, are often victims of LV. A large qualitative research study that contained three sub-studies exploring professional identity development in different professional stages was conducted by a research team in Macau, Special Administrative Region of China. Semi-structured interviews with nursing students and clinical nurses were carried out; among the 58 participants in the three sub-studies, 20 described some forms of LV and their ways of dealing with them. Framed by the feminist perspective, the researchers explored young nurses’ coping strategies in dealing with LV perpetrated by senior colleagues. Two themes were developed reflecting the coping strategies for LV: “making extra efforts” and “soothing emotional distress”. Three sub-themes were under the theme of “making extra efforts”: “catching up knowledge”, “making the most use of learning resources”, “adjusting communication manner”; another batch of sub-themes was under the theme of “soothing emotional distress”: “seeking support from schoolmates”, “living with family but crying alone”, and “adjusting lifestyle”. The study implied that young nurses exerted their agency in coping with LV in clinical practices. Nursing managers and educators should support young nurses’ efforts in overcoming power-based LV and incivility.

## 1. Introduction

The famous idiom “nurses eat their young” embodies lateral violence (LV) as a part of the nursing environment [1]. LV is a form of bullying defined as physical, emotional, or verbal abuse of an employee of the same rank or position. In nursing, LV is described as nurse-to-nurse aggression. It has destructive effects on victims both physiologically and psychologically, such as tiredness, sleep deprivation, depression, and lower self-esteem [2,3]. These can affect the professional performance of nurse victims, resulting in poor patient care quality. Bystander nurses who have witnessed LV may hold negative attitudes toward nurses and nursing and struggle to establish their professional identity [4]. Around 10–30% of the nurse victims of LV decided to resign from their position and 7–35% considered leaving the nursing profession [2], worsening the already serious problem of nursing shortage.

Despite the adverse effects of LV on nursing, it is often taken for granted by nurses. Nursing educators and managers are allegedly not sensitive to LV among nursing students and clinical nurses or do not care about it, even if they know it is happening [3,4]. Young nurses themselves see it as an inevitable path in starting their nursing careers. In addition to the indifference to LV in the clinical nursing arena, previous studies on LV in nursing have focused on the styles and consequences of LV [3,4]. There is limited empirical evidence on the nurse victims’ coping strategies to address LV. A critical analysis framed by theoretical perspective into nurse victims’ coping strategies may induce new insights into the dark side of dynamic interactions among nurses.

### 1.1. Prevalence of LV among Nursing and Victim’s Coping Strategies

Despite the bullying and uncivil practices in almost every healthcare setting, the prevalence of LV among nurses varies in different regions and healthcare institutes. One review reported an LV prevalence among nurses that ranged from 1% to 87.4%, and the most common manifestation is psychological harassment as opposed to physical aggression [2]. The review displayed a regional disproportion among the original studies, with most of the studies being conducted in Western countries [2], indicating further research is needed in other regions.

Studies on bullying in nursing in Asian regions are limited. In Hong Kong, about 7.7% of clinical nurses reported being bullied by their colleagues in workplaces, including physicians, nurses and other healthcare professionals [5]. One study in mainland China examined bullying in the Operation Room (OR) and found a prevalence of 15.8%, mainly from patients, patients’ relatives, and physicians [6]. There is a general lack of research exclusively on LV among nurses.

Scholars have summarized the bullying victims’ coping strategies in oppressed positions into three categories: emotion-focused strategies, problem-based strategies, and community-based strategies [7]. The emotion-based strategies emphasize denial and disengagement; problem-based strategies include compensation, skills training and self-efficacy; and community-based strategies mainly refer to social support.

Research on how nurses deal with bullying supports the coping measures mentioned above. One study articulated six ways of coping used by clinical nurses in Korea, including problem-focused strategies, wishful thinking, detachment, seeking social support, focusing on the positive, and tension reduction [8]. Another study on Turkish nurses revealed that a small number of nurses used retaliation as revenge on their bullies, but this was carried out secretly. Astonishingly, only some of the nurses recommended measures to deal with bullying, while most of them had no suggestion for the problem [9], which indicates the formidable nature of the problem.

### 1.2. Feminist Perspective and LV among Nurses

Theories can reshape ways to understand practices and provide a tool for change. Scholars have called for feminist theory to be applied in healthcare education as well as in health care practices [10,11]. Feminist theory is the extension of feminism into the theoretical discourse and it refers to a family of critical theories and approaches. The feminist perspective implies understanding interpersonal relationships through the lens of feminist theory. The study of postmodern feminism has extended gender inequality to the fundamental essence of power inequality, which exists not only among males and females but also among people of the same sex. Much research has focused on workplace bullying of nurses perpetrated by physicians or organizational managers who are predominantly males [5,6,12]. Attention should be paid to LV among nurses. Nurse managers, head nurses, and senior nurses are at the top of the hierarchal health organizations, whereas nursing students and newly graduated nurses are at the bottom. Those with less power are usually the victims of bullying and incivility.

This oppressed position made people politically apathetic and unwilling to change this stereotype [7,13]. It has been found that nurse victims of LV are unwilling to report their authorities’ bullying behaviors. Research framed by the feminist perspective may have elevated vigilance and awareness of the hidden nature of LV and incivility within nurse groups [14]. Previous research has been carried out on the personalities and behaviors of people who are prone to workplace violence victimizations [3,15]. For example, those with low self-esteem are likely to become victims of workplace violence [15] and those who are interpersonally demanding and difficult to get along with others tend to become targets of aggression from other organizational members [16]. The underlying perspective that some victims are to blame for their victimization may have legitimized workplace violence to a certain extent [17].

Feminist theorists believe that individuals have the ability to make a change. They value the subjectivities and experiences of the oppressed as legitimate knowledge from which change can take place. Framed by the feminist perspective, this study will provide a chance for young nurses to speak out about their sufferings as the victims of LV in clinical practices and their efforts to overcome difficulties and struggles. Their voices would be as powerful as those in the powerful positions in bringing about meaningful changes to tackle the dark sides of nursing practice.

## 2. Methods

### 2.1. The Study Context

A large qualitative research study that contained three sub-studies was carried out in Macau, Special Administrative Region (SAR) of China. As an SAR, Macau has a high degree of autonomy in governing society and has different economic, political, and social systems from those in mainland China. It is a small city with a population of 679,600 [18]. There are 2491 registered nurses in Macau, with 3.7 nurses per thousand of the population [18]. This figure is much smaller than those in other developed regions such as Hong Kong (7.6 in 2018), Singapore (6.2 in 2017), and Korea (7.3 in 2017) [19]. There are two nursing schools in Macau training pre-registered nurses—one private and the other public. The private nursing school is the larger one with a history of nearly 100 years and has contributed approximately 70% of the registered nurse workforce.

### 2.2. The Participants

The data for this paper are from the large study. Purposive sampling was used in the study to invite nursing students and alumni from the private nursing school. The sub-studies explored nurses’ professional identity development in their different career stages, from nursing students to clinical nurses with varying lengths of clinical practice. While the researchers were sensitive to the LV among nurses, not all the participants encountered LV. Most of the participants described positive experiences with their clinical supervisors and other senior nurses. The researchers identified the data of 20 participants who talked about some forms of LV.

### 2.3. Data Collection

The time and place of the interviewing were scheduled at the mutual convenience of the interviewees and the interviewers (the researchers). Semi-structured interviews were performed in each of the three sub-studies. Three members of the research team, A.M, H.L.T., and P.L.C., conducted the interviews, respectively. All four members of the team were registered nurses and faculty members of the nursing college. One of them was a male nurse, while the others were females. They were experienced qualitative researchers and were not involved in the evaluation of participants’ clinical performances. All the interviews happened in the quiet rooms of the nursing school and for each interview, no one else was present in the room except the interviewer and the interviewee. The interview questions were designed for the specific purposes of the individual sub-studies. Those questions related to interpersonal relations among nurses included: “What difficulties or challenges did you come across in terms of interpersonal relationships with other nurse colleagues? How did you deal with the difficulties or challengers?”, ”How do you feel about the atmosphere of the wards you worked?”, “Please tell me with whom you worked impressed you the most?”, “How about your future career plans?” Probing was used to obtain LV encounter details should the young nurses describe their relationship tensions with their supervisors or other nurses. One interview was conducted with each of the participants. Each of the sub-studies reached saturation for its specific research purpose. For example, the sub-study on the effects of female nurses’ clinical practices on reshaping professional identity had 33 participants when saturation appeared. The interviews were recorded, and permissions were obtained from all the participants for the recordings. Field notes were written by the interviewers immediately after interviewing, recording the contexts of interviewing and the overall impressions of the interviewers about the interviewees’ performances. The interviews lasted from 28 to 78 min, with most of them being within 60 min. The average time was 43 min.

### 2.4. Data Analysis

The interviews were transcribed verbatim. The researchers reviewed all the interviews in the three sub-studies. For example, in one of the sub-studies containing 33 participants, interviews from 13 participants mentioned LV and thus, were selected for data analysis for this paper. The other seven participants with LV experiences came from the other two sub-studies, with a total of 25 participants. NVivo 11 Plus was used to facilitate data analysis. Classic thematic analysis for qualitative research data was used [20], including four steps.
Familiarizing the data. The researchers read and re-read the interview transcripts, sometimes against the interview recordings, to obtain a comprehensive understanding of the participants’ experiences.Assigning sections of the interview text into meaningful units, labeling them with codes.Looking for connections between the codes and grouping the connected codes into sub-themes and sub-themes into themes.Formulating themes as the expression of the latent content of the interview texts.

### 2.5. Ethics Considerations

The study received approval from the research committee of the nursing school where the researchers worked (Reference No: 2016 JAN01). It was funded by the Macau Foundation (Reference No: 1962/DSDSC/2016), a government-sponsored organization to support the development of science and technology. The researchers explained the purposes and process of the study to the potential participants during the initial telephone contacts and opportunities were provided for clarification of any questions and concerns from the potential participants. The participants were assured that participation in the study was out of their willingness. Confidentiality and anonymity were guaranteed as a code number was assigned to each of the participants, and no personal information would be revealed beyond the research team. The participants were asked to read and sign the consent form before the interviews were carried out.

### 2.6. Trustworthiness of the Study

Both COREQ (Consolidated criteria for reporting qualitative research) [21] and principles advocated by Guba and Lincoln [22] to ensure the quality of qualitative research were used to guide the conduct of the study and thus, various measures were taken to enhance the quality of the study. Four members of the research team independently coded three interviews; they then compared and discussed the coding results. Consequently, a coding framework was established. The first author, A.M, then used the coding framework to code the rest of the interviews and the analytical results were discussed and reached agreement among the team members. Member checking was used by sending the analytical results back to a few of the nurse participants for their feedback. No inconsistency existed between the researchers and the nurses in terms of the emerged themes and sub-themes. Detailed descriptions were provided not only on the experiences and viewpoints of the nurses but also on the contexts of their experiences. An audit trail was performed by keeping a clear track of the research process. Similar backgrounds in nursing between the researchers and the participants had benefits for the researchers to perform data collection and data analysis. At the same time, the researchers were cautious about the possible bias brought in by their identity as “insiders” [22]. During the analytical phase, the researchers constantly reflected on their assumptions, preconceptions and values, and the possible impacts of these on the analytical results.

## 3. Findings

### 3.1. Characteristics of the Participants and Their LV Experiences

The LV victims were nursing students and junior nurses who had worked no more than three years, resulting in a cohort of young nurses. Four of them were males and the other 16 were females. Only one was married. The others were single and lived with their family members. A brief background of the 20 participants and their experiences with LV are listed in Table 1. Codes are used to conceal the participants’ identity, with NS referring to the nursing student, FN to female clinical nurse, and MN to male clinical nurse.

The students described their LV experiences during their 8-month clinical studies when they rotated in different clinical wards, working under a nurse supervisor. The clinical nurses alleged that their LV encounters took place mainly in the early stages of their new role as a clinical nurse, particularly in the first 6 months of their new employment, when they were in the probation period under a senior nurse’s supervision. All the young nurses, including the nursing students and novice nurses, had heard of the difficulties and challenges they might encounter before entering their new role. However, to some participants, the extent of the suffering they endured later was beyond their expectations. Generally, the perpetrators or the bullies were their supervisors, while other senior nurses sometimes bullied the young nurses too.

### 3.2. Coping Strategies Used by Young Nurses

Two themes emerged from the data, indicating the strategies used by young nurses in dealing with LV: making extra efforts and soothing emotional distress. There are sub-themes under each of the two themes. The sources and frequency of themes and sub-themes are presented in Table 2. A detailed description of the themes and sub-themes is provided in the following sections.

#### 3.2.1. Making Extra Efforts

Almost all the participants had tried their best to prevent LV from happening. The efforts they made included: “catching up knowledge”, “making the most use of learning resources”, and “adjusting communication manner”.

• Catching up knowledge

The participants described insufficient knowledge and skills in patient care, and their unawareness of the specific rituals and rules in the specific wards when they arrived at a ward, although they had had much clinical practice in hospitals in previous training. Inability to conduct the performances expected by their supervisors and other colleagues was the biggest trigger for the tensions and conflicts between the juniors and the seniors. Furthermore, due to the nursing workforce shortage, they were required to conduct tasks and responsibilities before they felt ready to. Despite feeling it was unfair that they were reprimanded for the work they were unable to do, they were determined to spend more time to learn more. One nurse, who had worked in the ICU for seven months since graduation, described her efforts to improve their knowledge and skills:
*We had to work on shifts on our own in two months; so there was much pressure on us. The first two or three months were tough because we had not worked on so many machines in the ICU. There were many patients with impairments of the central nervous system or cardiovascular diseases, and I had to learn nursing care related to those diseases. I used to study for about three hours every day after I went home from the hospital.*FN 23Other participants echoed that they were so busy with the new role that they had no time for anything else. One participant had to suspend her Master’s degree study in a university in Hong Kong after changing from a community clinic nurse to a hospital nurse.
*I did not work in a hospital after graduation because of my health problem. I was working in a private clinic, which was not very busy, so I attended a Master’s program in Hong Kong. After my health had improved, I left the clinic and went to this hospital. I knew that I would be reprimanded if I could not get familiar with my work quickly. I decided to suspend my Master’s study for one year to concentrate on my current job.*FN 22

• Making the most use of learning resources

Nursing students and newly graduated nurses were all assigned to one supervisor, and the participants spoke highly of the importance of the supervising role. Most of the participants talked about learning from their supervisors. However, when the junior–senior relationship became sour, the junior might change their learning style. One nursing student spoke of the strained relationship with her supervisor, and she learned by “stealing”.
*That was a tough time. I just went to a surgical ward. I was a talkative person, but I was silent when I was with her. We had no communication because it was so easy for her to get angry. I was scared by any word or action from her. Instead of asking her question, I quietly observed her. I can say that I stole her knowledge and skills.*NS2

Other participants also mentioned learning from other senior nurses because their supervisors were not always with them or the seniors were reluctant to teach their junior.
*I did not learn much from my supervisor, honestly. My supervisor was too busy to teach me. Then she took her annual leave. I was left alone. I observed what other nurses were doing and then went over, asking, “Can I have a look?”. They would allow me to watch and tell me the performance tips.*FN 25
*My supervisor and I did not like each other. I would not ask her questions. Some of the nurses were willing to teach me, and I would learn from them.*NS4

• Adjusting communication manner

The participants would reflect on communication manners when the unfavorable relationship with their supervisors or other colleagues arose. One nurse who had worked for seven months stated her attempts to change her way of communication with the head nurse.
*I am a straightforward person, and my head nurse did not like the way I dealt with things. I did not want to be disliked by her. Now I have changed a lot when I am in the ward, not that straightforward and speak more carefully. But when I have finished work, I am myself again, like what you see right now. I am happy and laugh and speak loudly.* (laughing)FN27

While this nurse stated difficulty in changing their personality, other participants confirmed that being a nurse helped them develop ways of dealing with people. They had become thoughtful and more emphatic when dealing with interpersonal relationships. As such, they had become more mutual and been more adept in communicating with the colleagues in their ward.

#### 3.2.2. Soothing Emotional Distress

Whereas the participants had tried their best to prevent conflicts and confrontations with their supervisors and other senior nurses, unhappy relationships did happen, and the young nurses sometimes experienced verbal and nonverbal harassment, humiliation, or other uncivil treatments from the seniors. “Soothing emotional distress” is a theme to reflect the emotional distress triggered by the participants’ LV experiences and their ways of dealing with unhappy feelings. Young nurses had sought various ways to regain psychological health to go on with their clinical practices.

• Seeking support from schoolmates

As there are only two nursing schools and three hospitals in Macau, the participants had many schoolmates in the same hospital or even in the same ward. They articulated that they managed to dine out with schoolmates, despite their tight learning and work schedules as new nurses. They would exchange learnings in patient care and share coping strategies. One nursing student talked about mutual support with her schoolmate.
*We still felt difficult, although we had practiced in this hospital over the previous three years. One of my high-school friends studied nursing and felt difficult too. We exchanged our difficult experiences and then supported each other, hoping that we two could go on with our journey. We encouraged each other to like our major and be happy again.*NS1

For the participants, their schoolmates were comforters for their emotional distress and information and learning sources.

• Living with family but crying alone

Macau is a small city, and all the young nurses but the married ones lived with their family members. However, few of them told their family of their encounters and difficulties in working. They figured that nursing care was unique and ordinary people might not understand nurses’ work. Furthermore, they did not want their family to worry about them.
*Because they *(family members)* are not in the health system, they don’t understand us. If I told them that I failed in IV (Intravenous injection), they would wonder, “Why were you not be able to do that? Isn’t it very simple?” They don’t know that IV is not that simple.*FN31
*This ward was what I chose to work in. But I was not happy in the first half-year, very unhappy! My colleagues were very strict with me. They did not care about my feelings and would accuse me under any circumstances. It was very stressful. My family could see that I was unhappy and asked why. I told them that I came across something difficult that new nurses would do, and I would be OK. But I was very upset. I used to close myself in my room and cry. I felt a little relief after I had cried.*MN4

• Adjusting lifestyle

A small number of the participants dealt with their stress by making healthy or unhealthy changes in lifestyle. One participant moved to another hospital after she had worked in her first hospital for nearly one year. She had worked in her new role for half a year. She described her pressure in her first workplace and how she coped:
*Not all the colleagues were kind to the newer. Some of them were harsh to us. I remember one time I dealt with one patient. His blood pressure was very low. The senior nurses were not happy with my dealings and then criticized my performance in this case and then extended it to my performance in other cases. It seemed that none of what I had done satisfied them. I was very stressed. Then I ate, ate, ate. I ate lots of desserts and chocolates, and I became fatter* (Both the participant and the interviewer laughing).FN34

One male nurse described his dealing with stress by drinking. He quit his position in the OR in a hospital after he had worked for eight months. He then moved to another hospital. He was so upset by his previous experience that he firmly refused to work in the OR again in his subsequent employment. He vividly described that he could not sleep without drinking because of unbearable distress:
*I drank beers every day. I could not sleep if I had not drunk. The next day when the clock alarmed me, I managed to get up and went to the hospital.*MN11

Another male nurse did not grieve by himself. He managed to go out and do some outdoor activities.
*That period was difficult. I found mates to go out. I did not only find the nursing schoolmates but also my high schoolmates. On weekends or holidays, several of us went out to climb hills or did other exciting activities. Going outside did not solve the problems, but you may feel refreshed. After all, you need to get over the difficulties and stand up again.*MN7

## 4. Discussion

This study’s findings confirmed the power-based disadvantaged status of nursing students and junior nurses prone to LV. Framed by the feminist perspective, the study reported that young nurses refused to stay in the victimized position and struggled to overcome their adverse conditions.

There is a gap between the theoretical knowledge taught in the classroom and the clinical practices in healthcare settings. Our study showed that this gap was further complicated by the nursing workforce shortage, exerting pressure on young nurses and becoming the essential trigger for LV and incivility among nurses. The problem-focused strategy of making extra efforts can enhance the young nurses’ self-efficacy in performing their new roles. This strategy is widely reported by other studies [7,23,24], reflecting nurse professionals’ pragmatic nature.

Senior nurses are the model of professional identity for nursing students and junior nurses. This study showed that young nurses cherished their supervisors’ knowledge source, even if the relationship between the two became sour. Contrary to findings in other studies that devaluing people of lower status may induce low-esteem and discourage their learning motivations [7,25], our study showed the young nurses’ endeavor to subvert the negative impacts of the bullies by consolidating with other senior nurses. In this way, young nurses can still gain support from other nurse group members if they refuse to submit to the bullies’ influence. The young nurses in our study also exhibited their flexibility in interpersonal interactions. Implicit in the flexibility is that the young nurses become mature from the adverse experiences in clinical work and are increasingly adept in dealing with work-related personal interactions. Other studies supported that nurses can reframe burdensome experiences and adjust frames for learning [26].

Social support is the most used coping strategy in bullying and incivility because it provides assurance, creating a sense of relaxation and security, lifting spirits, and emotional belonging and involvement. Studies showed that for nursing students, their family members and friends were their most important source of social support [27], whereas for new nurses, peer nurses and particularly the experienced nurses were an important source [28,29]. Usually, newly graduated nurses will not have continued access to their schoolmates, teachers, and friends after going to their new workplace. They may feel lonely and isolated, so support and acceptance from peer nurses are important. Our study showed social support as an important emotion-focused strategy [10], but with cultural and regional characteristics. Macau is a small and close-knit society. The young nurses’ friends studying in the same profession often become colleagues in the future. This specific phenomenon confirms the importance of peer support as well. However, it is alarming that some young nurses isolated themselves from social life because of their clinical studies [30]. As a result, social support could not be delivered.

Emotional eating, in which eating is not to satisfy feelings of hunger but to relieve stress, is a typical strategy to deal with stress. Other studies found that overeating happened with those under pressure, and the consumption of fast food and snacks was higher than vegetables and fruits [31,32]. The fact that very few of the participants in our study overate might be attributed to their professional knowledge of overeating’s adverse effects. Exercising can relieve stress and is beneficial for physiological fitness [33]; however, only a few young nurses in our study exercised regularly.

### 4.1. Implications for Nursing Management and Nursing Education

Framed by the feminist perspective, this study showed young nurses’ efforts to get themselves out of hostile treatment against them. However, implicit in the young nurses’ experiences was that none of them reported their unfair treatment to the authorities, an indication that the young nurses were lonely and quiet fighters in combating power-based inequality within nursing. They deserve more support from people around them.

A systematic review found an overall positive impact of support on young nurses’ wellbeing during their transition period, regardless of the type of support provided [34]. Nurses in a higher power position, such as nurse managers, head nurses, and senior nurses, should not ignore the vulnerability of young nurses to bullying activities and should provide any kind of support they can, such as implementing a zero-tolerance policy to bullying, reasonably assigning tasks to young nurses, making it easy for young nurses to speak out about their victimization, modeling good behaviors to prevent bad behaviors, etc. Nursing codes decree that nurses should treat all patients with dignity regardless of their backgrounds. Nurses in different power positions should extend such kind of treatment to their colleagues. Our study revealed that even head nurses committed uncivil behaviors towards novice nurses. Nurses as a whole need to implement nursing codes without slack. In this way, nurses of different classes, races, and ethnicity can eliminate power-based inequality between nurses and create a harmonious working environment to nurture young nurses.

The study showed that supervisors are the most important learning resources for young nurses and the largest source of stress as well. Supervisors for new nurses should be carefully chosen from clinical nurses. Young professionals hoped that their supervisors would provide illumination and help with emotion management [35]. This means that while professional competency is important for supervising, it is more important that supervisors be emphatic to young professionals. Research has proved the effectiveness of empathy interventions in preventing violence and uncivil behaviors [36,37]. Empathy interventions may include role-playing, watching videos or films related to violence and suffering, viewpoint challenging between victims and perpetrators, etc. Such interventions involve not only victims but also perpetrators and can positively change the profiles of the perpetrators, including enhanced communication skills, reduced jealousy, and improved ability to understand the emotional status of others. Few empathy interventions have been conducted in nursing, indicating future research directions, because empathy interventions can enhance nurses’ perspective-taking trait. Additionally, this study showed the strained relationship between junior and senior nurses further complicated by nursing workforce shortage. The nursing shortage is a challenging problem that nursing managers and policymakers should address [38].

Nursing educators have their part in combatting bullying and incivility in clinical practices. Feminist pedagogy has been adopted in nursing education, in which a student-centered model has been established [39,40]. However, nursing educators have a responsibility to extend the feminist perspective beyond the classroom and into clinical settings. They can enhance students’ awareness of power and identity-based inequality in the clinical world. Scholars have suggested a multi-component progressive educational program in nursing schools which includes understanding different styles of bullying, consequences, prevention and coping strategies. Such a program can be incorporated into the curriculum for students from sophomore to senior year [41]. This is in accordance with nursing students’ suggestions for early education on bullying and how to cope in school education [42,43]. Our study showed experience sharing among peers to deal with bullying and incivility. Nursing schools can organize workshops, in which final-year nursing students or newly graduated nurses share their anti-bullying experiences with junior schoolmates.

### 4.2. Limitations of the Study

While the data for this paper came from several sub-studies that involved both nursing students and junior nurses, the participants were either students or alumni from one of the two nursing schools in Macau. The non-randomized sampling with the small sample size limited the study’s strength to generalize its findings [44]. Additionally, victims of destructive or harassment activities tend to hide their sufferings from other people [8,23,45], and this might prevent the participants in our study from revealing their stories openly.

## 5. Conclusions

The study explored the coping strategies applied by nursing students and junior nurses to deal with LV in clinical practices. Using the feminist perspective, the study provided an opportunity for the young nurses, who are usually silenced by nursing’s hierarchical power system, to speak out about their victimized experiences. It revealed that young nurses made extra efforts to prevent LV from happening, and once LV happened, they took various measures to soothe emotional distress. Young nurses were not passive receivers of uncivil practices but active players in combatting bullying and other uncivil activities. While young nurses’ coping strategies echoed findings in other studies, they were framed by local social characteristics. Hidden in their story to deal with a bully is that young nurses fight bullying by themselves. This calls for the active involvement of nursing managers and nursing educators to support young nurses’ endeavors to prevent bullying from happening or minimize its effects on psychological wellbeing.

## Figures and Tables

**Table 1 ijerph-18-07167-t001:** A brief background of the 20 participants and their LV experiences.

Code	Age	Work Experiences	LV Experiences
NS1	21	Just finished the bachelor’s nursing program	Ignored by supervisor Criticized by the supervisor for not being active in clinical learning
NS2	22	Just finished the bachelor’s nursing program	Scared by each word and each action of her supervisor Informed by supervisor of her disappointment with the junior
NS4	22	Just finished the bachelor’s nursing program	Having a too heavy workload Expected by supervisor to have known everything in patient care
FN22	26	Worked in private clinics for 14 months and in the Emergent Department (ED) of a hospital for three months	Fear of being reprimanded by senior colleagues Embarrassed when asking questions
FN23	23	Worked in the Intensive Care Unit (ICU) of a hospital for seven months	Colleagues not friendly Having a too heavy workload
FN24	22	Worked in an adult nursing ward of a hospital for seven months	Having a too heavy workload Frequent scrutiny from senior colleagues
FN25	23	Worked in the Pediatric Ward of a hospital for one year and a VIP Pediatric and Maternal Ward for eight months	Supervisors in different work shifts from her
FN26	23	Worked in the Operation Room (OR) of a hospital for seven months	Reprimanded by supervisors for not quick enough in doing things Witnessing one of the new nurses leave his job
FN27	22	Worked in a VIP surgical nursing ward of a hospital for seven months	No support from supervisors when being refused services by patients Supervisor not satisfied with her manner of communication
FN28	26	Worked in a VIP adult nursing ward in one hospital for two years and the general adult nursing ward in another hospital for one year	No support from supervisors and colleagues when having conflicts with patients Reprimanded by colleagues when not writing medical records correctly
FN29	27	Worked in a surgical nursing ward of a hospital for one year and the Pediatric and Maternal Ward for another year	Frequent scrutiny from senior colleagues No support from supervisors and colleagues when having conflicts with patients Felt unprepared when being put on duties independently
FN30	25	Worked in the Newborn Care Ward of a hospital for one year and the Newborn and Material Ward in another hospital for two years	Minor errors were amplified Ignored by supervisors and other senior colleagues
FN31	23	Worked in the VIP Pediatric Ward of a hospital for seven months	No instruction from supervisors
FN32	22	Worked in the ED of a hospital for seven months	Having a too heavy workload Embarrassed when asking simple or repeated questions
FN33	28	Worked in an adult nursing ward in a hospital for one year and the Hemodialysis Room (HDR) in another hospital for half a year	Ignored by colleagues Embarrassed if not asking appropriate questions
FN34	28	Worked in a surgical ward of a hospital for one year, in a school clinic for one year, and in an adult nursing ward of the second hospital for half a year	Reprimanded by supervisor if asking a question for a second time Expected by supervisor to have known everything in patient care Supervisor was harsh to her Senior nurses were too busy to teach her anything
MN4	26	Worked in the ED of a hospital for nearly two years	Ignored by colleagues Criticized by the supervisor when other nurses were present
MN7	26	Worked in the adult nursing ward of a hospital for nearly two years	Having a too heavy workload Required to conduct tasks beyond capability
MN11	27	Worked in the OR of a hospital for eight months and HDR in another hospital for two months	Reprimanded by supervisor every day Ignored by senior colleagues Supervisor spread gossip about him Probation tests failed by supervisor
MN18	27	Worked in the ED of a hospital for three years	Reprimanded by supervisor at present of other colleagues

**Table 2 ijerph-18-07167-t002:** Coping strategies used by the young nurses to deal with LV.

Themes/Sub-Themes	Number of Participants	References
Making extra efforts		
Catching up knowledge	6	[13]
Making most use of learning resources	6	[10]
Adjusting communication manner	2	[3]
Soothing emotional distress		
Seeking support from schoolmates	10	[17]
Living with family but crying alone	6	[8]
Adjusting lifestyle	3	[4]

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
