# Peer review of "“You Need to Get Over the Difficulties and Stand Up Again”—A Qualitative Inquiry into Young Nurses’ Coping with Lateral Violence from the Feminist Perspective"

_ijerph, 2021, doi:10.3390/ijerph18137167_

Round 1
Reviewer 1 Report
Please, check page 2, line 89-91: These sentences are quite unclear.
The 1.2 paragraph on Feminist perspective and LV among nurses is not well focused. It seems to be more focused on mis-management of the workplace. It si not clear how the feminist perspective is offering an insight or a due perspective of the issue. Please, make this point more explicit and make also more clear how this perspective is different from a management perspective of organizational well-being and safety; This is crucial pint to me, since several of the negative organizational behaviors reported by the nurses are more related to bad people management than to an anti-feminist attitude. Besides, it seems that real LV was perpetrated by other woman, and not by male colleagues.
Finally, in the Discussion paragraph, author refer to "bullying and other uncivil behaviors", while e these constructs are not introduced before
More information and details are needed about the semi-structured interview that was administered.
Which areas were exactly explored?
Which were the question posed?
Interviews views lasted from 28-78 minutes; it seems to me that 78 minutes is rather a quite long and demanding period. What they were so long? I ask this because long interviews are demanding and tiring, both for the interviewer and the interview, and hold the risk of being not fully reliable, in the end.
Reviewer 2 Report
-Most of the literature is rather old, I would suggest to update it
-Please, justify sample size
-The sample should be better characterised
-More information is needed in how the agreement between researches was reached and its statistics.
-the limitations only are focused on the qualitative methodology but not on the study itself.
Minor:
"The famous idiot “nurses eat their young’ embodies lateral violence (LV) a" Could be reformulated?
-There is no need in the code in table 1
Round 2
Reviewer 2 Report
I thank authors to address most of my suggestions.
I am afraid that two points remain not satisfactory:
- Methodology. Authors need to justify it. Particularly aspects related to sample size. This is a well reputated journal and they would like to be cited in the future. Please, address this issue for future replications.
- Why these results are important in an applied level? What is the take.home message? It is not enough to state "nurses shouldn’t ignore the vulnerability" From my point of view empathy intervention might be the key. These are very commond in other profiles (abusers such as the following cite: Van Hoey, J., Moret-Tatay, C., Santolaya Prego de Oliver, J. A., & Beneyto-Arrojo, M. J. (2019). Profile changes in male partner abuser after an intervention program in gender-based violence. International journal of offender therapy and comparative criminology, 0306624X19884170.), but not so popular in nursing, as they are supposed to have this variable trained. This merits to be explained.
